# Insect Protein-Based Diet as Potential Risk of Allergy in Dogs

**DOI:** 10.3390/ani11071942

**Published:** 2021-06-29

**Authors:** Blanka Premrov Bajuk, Petra Zrimšek, Tina Kotnik, Adrijana Leonardi, Igor Križaj, Breda Jakovac Strajn

**Affiliations:** 1Institute of Preclinical Sciences, Veterinary Faculty, University of Ljubljana, Gerbičeva 60, 1000 Ljubljana, Slovenia; petra.zrimsek@vf.uni-lj.si; 2Small Animal Clinic, Veterinary Faculty, University of Ljubljana, Gerbičeva 60, 1000 Ljubljana, Slovenia; tina.kotnik@vf.uni-lj.si; 3Department of Molecular and Biomedical Sciences, Jožef Stefan Institute, Jamova 39, 1000 Ljubljana, Slovenia; adrijana.leonardi@ijs.si (A.L.); igor.krizaj@ijs.si (I.K.); 4Institute of Food Safety, Feed and Environment, Veterinary Faculty, University of Ljubljana, Gerbičeva 60, 1000 Ljubljana, Slovenia; breda.jakovacstrajn@vf.uni-lj.si

**Keywords:** dogs, allergy, mites, mealworm, proteomics

## Abstract

**Simple Summary:**

There is growing interest in the use of edible insects as an alternative source of protein and fat in food/feed formulations. Yellow mealworm (*Tenebrio molitor*) larvae are rich in protein and other nutrients. Since insects are related to mites, a common allergenic species in dogs, we investigated the interaction between mealworm proteins and the immune system of allergic dogs sensitised to storage mites in this study. Using Western blot analysis, we confirmed the binding of IgEs from canine sera to mealworm proteins. With mass spectrometry analysis, we identified several *T*. *molitor* proteins, which are known as human allergens. The results of our study raised the possibility that dogs allergic to mites clinically show cross-reactivity to mealworm proteins.

**Abstract:**

Before insects can be used widely as an alternative source of dietary protein, their allerginicity should be investigated. Therefore, the aim of our study was to assess the potential adverse reactions of the immune system of dogs against *Tenebrio molitor* proteins. Dogs sensitised to storage mites *T. putrescentiae* and *A. siro* were included. Clinically healthy and clinically allergic dogs were compared. Proteins were extracted from mealworm larvae and their digestibility determined by in vitro incubation with digestive proteases. Mealworm protein extracts and digests were analysed by SDS–PAGE. Canine sera tested for the presence of mite-specific IgEs were used for subsequent Western blotting. LC-MS/MS analysis was used to identify mealworm proteins and their allergenic potential was predicted with the AllermatchTM tool. The binding of canine sera IgEs to mealworm proteins was confirmed; however, the differences between the two groups of dogs were not significant. Moreover, no clear correlation was found between sensitisation to storage mites and clinical status of the dogs. Altogether, 17 different proteins were identified, including tropomyosin, α-amylase, and Tm-E1a cuticular protein that are known cross-reacting IgE-binding allergens. Our results suggest that dogs allergic to mites may clinically express also the cross-reactivity with mealworm proteins.

## 1. Introduction

The increasing global need to find sustainable alternative protein/energy sources for animal nutrition has stimulated research in the field of non-conventional feed ingredients. Insects have great potential for several reasons: (i) their nutritional value, (ii) their feed conversion efficiency, (iii) the small space required to cultivate, and (iv), as they are nearly omnivores, they can grow on different substrates [1,2]. Insect proteins have nutritional advantages in total protein content and/or essential amino acid profile over plant proteins, e.g., cereals, beans, lentils or soybeans. They may also have advantages over animal meat due to their high content of high-quality protein (~50–85%) and significant amounts of other nutrients such as vitamins, minerals and lipids, including omega-3 and omega-6 fatty acids in favourable ratios. The proteins in edible insects also have high digestibility (up to 75–98%) compared to other protein sources [3]. It is, therefore, expected that insects will increasingly be used as a substitute for food and feed as a source of protein and energy [1,2].

There are already about 1500–2000 insect species and other invertebrates that are consumed by humans [3]. The larvae of the yellow mealworm (*Tenebrio molitor*, *T. molitor*), together with some other species, are among the most promising insects for integration into the European food and feed industry, which is due to the existing large-scale breeding know-how and the promising composition in terms of protein and fat content [4]. They have the ability to recycle low-value organic material such as organic waste and convert it into a protein source. The yellow mealworm is an insect species with one of the highest contents of proteins (47.8 to 53.1%) and lipids (27.3 to 38.3%) in dry matter, with energy contributions varying between 379 and 573 kcal/100 g [5]. Compared to beef, mealworm larvae also have significantly higher contents of linoleic acid, essential and non-essential amino acids, most markedly isoleucine, leucine, valine, tyrosine, alanine, and vitamins [3].

However, before insects can become an essential part of the diet, the potential risk of adverse food reactions (AFRs), including allergic reactions, should be investigated [6]. Allergy is a hypersensitivity reaction initiated by specific immunological mechanisms. It can be antibody mediated or cell mediated [7]. Allergic reactions to food are defined as adverse reactions to an otherwise harmless food or food constituent that causes an abnormal response of the body’s immune system particularly to one or more specific proteins in food [8]. The term IgE-mediated food allergy is used when IgEs are involved in the immunologic reaction [7]. Clinical manifestations of food allergy in humans can affect various organs and systems, including the skin, intestines, respiratory system, cardiovascular system and nervous system. Furthermore, clinical manifestations are highly variable, ranging from mild and localized manifestations of hypersensitivity, such as itching in the mouth, to severe, systemic, and often fatal reactions, such as anaphylactic shock [9]. The prevalence of food allergies in humans in Europe is up to 5.7% depending on age. Fish and seafood, peanuts and nuts, and fruit and vegetables are among the most allergenic foods in humans [6,9]. While anaphylactic reactions appear to be quite common in food-allergic humans, they are extremely rare in food-allergic dogs. Angioedema of the skin, vomiting, diarrhoea, conjunctivitis, and respiratory distress were documented in two dogs after nut ingestion [10,11]. Food allergies commonly cause skin diseases in dogs and cats. Their prevalence is estimated at about 5% of all skin diseases and up to 25% of all allergic skin conditions in dogs [12]. Cutaneous manifestations in food-allergic dogs often include pruritic, erythematous dermatitis of the face, ear canals, axillae, groin, and paws that is clinically indistinguishable from canine atopic dermatitis (AD). In any case of AD, where clinical signs are present throughout the year, food allergy can only be ruled out by strict elimination dietary testing. The presence of gastrointestinal signs, such as diarrhoea, vomiting, tenesmus, soft stools, flatulence, and an increased number of bowel movements is typically seen in food-induced canine AD. Respiratory symptoms may also be present [13].

Allergy following ingestion of insects can be due to the primary sensitization or cross-reactivity with another allergen [8]. Cross-reactivity is an immune-mediated phenomenon in which IgE antibodies recognize and bind similar allergenic molecules and trigger the immune response. IgE cross-reactivity often occurs between allergens in closely related species or evolutionary highly conserved molecules present in different species. Such molecules are known as pan-allergens. Cross-reactivity is important for several reasons, including the risk of allergic cross-reactivity to novel foods [14].

The yellow mealworm larvae (*T. molitor*) contain a variety of proteins, including proteins involved in metabolic functions, such as enzymes, and proteins with structural functions, such as muscle proteins. Among them, the IgE-binding cross-reacting allergens tropomyosin, α-amylase, arginine kinase and hexamerin have been identified as the most important food allergens in humans [15]. The phylogenetic tree classifies insects as part of the Arthropoda phylum, which makes them closely related to other allergenic species such as shrimps, prawns, cockroaches and house dust mites (HDM) [8]. In addition to house dust mites, another important group of mites is known as ‘storage mites’. They live in stored food and grain, but can also be found in kitchen floor dust, cupboards and pantries. Therefore, the term ‘domestic mites’ is used for all mite species that occur in the domestic environment and are able to trigger IgE-mediated sensitization. *Tyrophagus putrescentiae* (*T. putrescentiae*), *Acarus siro* (*A. siro*) and *Lepidoglyphus destructor* (*L. destructor*) are the main pest mite species on stored products. Several allergens of storage mites have been characterized and some of them are known as pan-allergens. Some of the allergens of *T. putrescentiae* show high IgE reactivity in vitro and in vivo [16].

In dogs, atopic dermatitis (AD) is a common chronic, recurrent, inflammatory and pruritic allergic skin disease, often associated with the production of IgE antibodies against environmental and/or food allergens. It affects 3–15% of the dog population. House dust mite allergens are the most common allergens recognized by the circulating IgE of atopic dogs, although there is cross-reactivity between house dust mite and storage mite allergens [17].

Mealworm proteins are now commercially available in dog feed formulations. Cross-reactivity and/or co-sensitization of various insect proteins has been demonstrated in humans sensitized to house dust mite and seafood allergens. In light of this research, the aim of the present study was to assess the potential cross-reactivity to mealworm proteins in dogs sensitized to the storage mites *T. putrescentiae* and *A. siro*.

## 2. Materials and Methods

### 2.1. Canine Sera

Thirty-one client-owned dogs of different breeds were included in the study. The group included 14 males and 17 females, aged between 0.8 and 7.8 years. The dogs were fed on commercial food and had no history of eating mealworm-containing foods. Medical history and clinical examinations (with the consent of the dog owners) were performed at the small animal clinic of Veterinary Faculty, University of Ljubljana, Slovenia. According to the results of clinical assessment, dogs were divided into two groups: clinically healthy (CH; *n* = 10, 2 males, 8 females, mean age 3.2 years) and clinically allergic (CA; *n* = 21, 12 males, 9 females, mean age 3.8 years). Blood samples were obtained by vein puncture of the jugular or cephalic vein and placed in tubes without anticoagulant. After 30 min of incubation at room temperature, samples were centrifuged, sera were collected, aliquoted and frozen at −20 °C. Batches of serum samples were sent to Alergovet S. L. (Spain) for IgE ELISA testing against 30 environmental- and feed-specific allergens. Based on the breakpoint values of the allergen-specific IgEs against two food mites *T. putrescentiae* and *A. siro*, the sera were further divided into the groups: IgE positive (IgEpos; *n* = 22) or IgE negative (IgEneg; *n* = 8). One sample gave borderline result.

### 2.2. Mealworm Proteins Preparation and Characterization

Raw yellow mealworms in final larval stage were bought at a local pet’s store and were kept frozen at −20 °C. After thawing, they were boiled for 30 sec and dried overnight in a thermal oven at 60 °C. Dried larvae were first ground in a mortar and then crushed with a grinder. The mealworm flour obtained was defatted overnight with petroleum ether (1:1 (*w*:*v*), Merck, Darmstadt, Germany) and dried at 60 °C for 2 h. For the acidic protein extraction, the mealworm flour was mixed with bidistilled water and the pH of the mixture was adjusted to 2 with 6.0 M HCl. After homogenization on ice using Ultra-turrax (5 × 30 s), the mixture was incubated for 1 h in a water bath at 40 °C and vortexed every 5 min. After 30 min of centrifugation at 8000 rpm and 23 °C the supernatant was recovered. Bidistilled water was added to the insoluble residue and the same procedure was repeated. The supernatants were combined, centrifuged again and concentrated. For the alkaline extraction of proteins, the procedure was identical, only the pH was adjusted to 10 with 4.0 M NaOH. Protein concentration was quantified using DC Protein Assay (Bio-Rad, München, Germany) and bovine serum albumin to plot a standard curve. The absorbance was read at 750 nm. A linear regression equation was used to determine the total protein concentration in the mealworm extracts. The samples were stored at −20 °C before further use.

#### 2.2.1. Digestibility of Mealworm Proteins

A total of 5 mL of acidic or alkaline mealworm extracts was diluted with 100 mM Tris-HCl to reach final protein concentration of 5 mg/mL and pH 2. Pepsin (Sigma-Aldrich, Steinheim, Germany) was added at a final concentration of 64.3 nM. Aliquots of 100 μL were taken at time 0 and after 60 min of digestion. After 1 h at 37 °C the pH of the digest was adjusted to 8.3 and trypsin (Sigma-Aldrich, Steinheim, Germany) at a final concentration of 87.1 nM and α-chymotrypsin (Sigma-Aldrich, Steinheim, Germany) at a final concentration 83.6 nM were added for further overnight digestion. Aliquots were taken for sodium dodecyl sulfate polyacrylamide gel electrophoresis (SDS-PAGE) analysis after the extraction procedure and before, during and after digestion.

#### 2.2.2. SDS-PAGE of Mealworm Proteins

For SDS-PAGE analyses, the alkaline and acidic mealworm protein extracts and digests were diluted 1:1 with Laemmli buffer (90 mM Tris-HCl pH 6.8, 4% (*w*/*v*) SDS, 20% (*v*/*v*) glycerol, 0.04% (*w*/*v*) bromphenol blue) with or without 10% β-mercaptoethanol, boiled for 5 min at 95 °C and separated on 12% acrylamide/Tris-HCl gels (MiniProtean, Bio-Rad, Hercules, CA, USA). Molecular mass markers (10–170 kDa; Thermo Fisher Scientific, Vilnius, Lithuania) were run in parallel to the samples. Following electrophoresis, the gels were either stained using Coomassie Brilliant Blue R-250 stain (Bio-Rad, München, Germany) to visualize protein components, or used for Western blotting.

### 2.3. Western Blot with Canine Sera

Proteins in mealworm extracts or digests were transferred from the gel to the PVDF membrane (Bio-Rad, Hercules, CA, USA). Blotting was performed at 200 V for 1 h in Tris/glycine transfer buffer containing 10% (*v/v*) methanol. The membranes were blocked with 5% (*v*/*v*) PVP (Polyvinylpyrrolidone; Sigma-Aldrich, Steinheim, Germany) in PBS buffer for 2 h at room temperature and then incubated overnight with dog sera (diluted 1:50) in PBS and 2.0% (*v*/*v*) Tween 20 (TBST) at 4 °C. Bound IgE were detected with HRP-conjugated goat anti-canine IgE (Novus Biologicals, Centennial, CO, USA), diluted 1:1500 in PBST and visualised with 3-amino-9-ethyl-carbazole (Sigma-Aldrich, Steinheim, Germany).

#### Statistical Analysis

The relationship between IgE-positive and -negative groups, according to the clinical status of the dogs, the presence of IgEs against selected storage mites and the occurrence of immunogenic bands, was evaluated using the exact Fisher test (Fisher Exact Test Calculator). *p* < 0.05 was considered significant.

### 2.4. Allergen Identification

The mealworm proteins were identified by liquid chromatography coupled with tandem mass spectrometry analysis (LC-MS/MS) and their potential allergenicity was determined.

#### 2.4.1. Mass Spectrometry Analysis

SDS-PAGE analysis of the mealworm protein digests was performed as described in Section 2.2.2. Proteins in gels were silver stained, each band was manually excised and de-stained [18]. Gel pieces were dehydrated with 100% acetonitrile (ACN) and subjected to in-gel digestion protocol with trypsin. First, the dried gel pieces were reduced and alkylated in one step with TCEP/CAA solution (10 mM tris(2-carboxyethyl)phosphine/40 mM chloroacetamide in NH_4_HCO_3_) for 30 min in the dark at room temperature. Gel pieces were then washed with 25 mM NH_4_HCO_3_, dehydrated with ACN and completely dried. Proteins were digested in-gel with 12.5 ng/μL MS grade modified trypsin (Sigma-Aldrich, St. Louis, MO, USA) in 25 mM NH_4_HCO_3_ at 37 °C overnight. Prior LC-MS/MS analysis the extracted peptides were purified with C18 StageTips prepared in house.

The MS analyses were performed on an ion trap mass spectrometer 1200 series HPLC-Chip-LC/MSD Trap XCT Ultra (Agilent Technologies, Waldbronn, Germany) [19]. The peptides were loaded onto the enrichment column in 95% (*v*/*v*) solvent A (0.1% (*v*/*v*) formic acid in water) and 5% (*v*/*v*) solvent B (0.1% (*v*/*v*) formic acid in ACN) at 4 μL/min, and eluted from the analytical column with a gradient of 5 to 50% (*v*/*v*) solvent B in 41 min, followed by a steep gradient to 90% (*v*/*v*) solvent B in 1 min, at a flow rate of 0.35 μL/min. MS acquisitions were carried out from 400 to 2200 m/z, followed by MS/MS scans of the five most abundant ions in each MS scan.

The MS spectral data were searched using the Spectrum Mill software Rev A.03.03.084 (Agilent Technologies, Santa Clara, CA, USA) against the *Tenebrioninae* species sequence database (4178 entries) extracted from the non-redundant NCBI (National Centre for Biotechnology Information) protein databank in November 2019. The following parameters were used: two missed cleavages were allowed, peptide charges +2 and +3, peptide and fragment mass tolerance of ±2.5 and ±0.7 Da, carboxyamidomethylcysteine (C) as fixed modification and oxidized methionine as variable. The results were additionally validated using Scaffold 2 software (version 2, Proteome Software, Portland, OR, USA) with the following thresholds: protein confidence of 95% and one peptide per protein at 95% confidence. Proteins were identified at 0.6% Prophet false discovery rate (FDR).

#### 2.4.2. Prediction of Allergenicity

The allergenic potential of proteins in the mealworm extracts identified with MS was tested using Allermatch^TM^ tool [20]. The complete or partial protein sequences were searched against the AllergenDB original sequences database using an 80 amino acid sliding window alignment with a 35% cut-off percentage.

## 3. Results

### 3.1. Mite-Specific IgE Antibodies in Canine Sera

The results of Alergovet ELISA screening test to determine the presence of mite-specific IgE antibodies are collected in Table 1. The sera of 10 CH dogs and 21 CA dogs were analysed. The enviromental panel included testing against house dust mites *Dermatophagoides pteronyssinus* (*D. pteronyssinus*) and *D. farinae* and against storage mites *A. siro*, *T. putrescentiae* and *L. destructor*. Storage mites *A. siro* and *T. putrescentiae* were selected as important for further discussion.

IgE antibodies against *T. putrescentiae* and *A. siro* were detected in the group of dogs with allergic symptoms in 66.7% and in clinically healthy group of dogs even in 80.0%. One serum contained borderline values and was excluded from the statistical analyses. We observed no correlation between sensitisation to storage mites (*A. siro* and *T. putrescentiae*) and clinical signs of allergy (*p* > 0.05). Majority of sera (93.5%) did not contain IgEs against the house dust mite *D. pteronyssinus*, but 77.4% of dog sera contained IgEs against *D. farinae*. Only one serum from clinically healthy dogs had IgE against *L. destructor* (data not shown).

### 3.2. Isolation of Mealworm Proteins

Raw and frozen *T. molitor* larvae were ground, defatted and the proteins were extracted at acidic (2) and alkaline (10) pH. The SDS-PAGE analysis of the extracts (Figure 1) showed several bands in a molecular mass range from 10 to 72 kDa. The acidic extract contained a larger number of proteins than the alkaline extract. Only a few weak protein bands of ~21 to 55 kDa were observed in the latter case. The SDS-PAGE protein patterns of acidic or alkaline extract were very similar no matter if analysed under reducing or non-reducing conditions.

### 3.3. Digestibility of Mealworm Proteins

The digestibility of the isolated mealworm proteins was demonstrated by time-dependent in vitro incubation of the extracted proteins with different digestive proteases: pepsin, trypsin and chymotrypsin. The resulting digests were analysed by SDS-PAGE to observe the extent of digestion (Figure 2). Protein patterns of acidic (pH 2) and alkaline (pH 10) mealworm extract digests are very similar. Most degradable proteins were hydrolysed already within the first hour with pepsin. Proteins or their parts that resisted an additional, very long (18 h) incubation with trypsin and α-chymotrypsin appeared at ~10, 15, 20, 30 and 43 kDa. To identify them, the respective bands were excised from the gel and analysed by MS.

### 3.4. Cross-Reactivity of Mealworm Extract Proteins and Canine Sera IgEs

We compared the immune cross-reactivity of mealworm extracts and their digests with sera of CA and CH dogs. The binding of IgE antibodies from canine sera to mealworm proteins was assessed using Western blot analysis (Figure 3). IgEs from all tested dog sera strongly cross-reacted with several proteins of 20–30 kDa in the non-digested mealworm acidic extract (Figure 3; black arrow). The mealworm proteins of 34–55 kDa (Figure 3; upper orange arrow) were recognized however by only ~63% of canine serum IgEs (Table 2; 19 of 30 sera). Canine IgEs identically recognized proteins in non-reduced and reduced mealworm extracts. In the fully digested mealworm acidic extract (1 h pepsin followed by 18 h trypsin and α-chymotrypsin digestion; PTC_18h_) only one protein band with ~14 kDa, remained weakly cross-reactive with canine serum IgEs (Figure 3; lower orange arrow). Again, difference in the binding of sera from the CH and CA groups to this protein were observed (Table 2); however, the difference was not significant (Table 2; *p* > 0.05). Interestingly, the binding of canine IgEs to mealworm protein alkaline extract and its digest was negligible (Figure 3).

Furthermore, we investigated the correlation between clinical signs of allergy in dogs, the presence of IgEs against the storage mites (*T. putrescentiae* and *A. siro*) in their sera and the cross-reactivity of sera with the proteins of mealworm protein acidic extract (Table 2).

When comparing the two groups of dogs (CH and CA), cross-reactivity of canine sera with mealworm proteins of 34–55 kDa was observed in the case of 75% of the mite-specific IgE-positive CH dogs and at ~57% of the mite-specific IgE-positive CA dogs. The percentage of cross-reactivity with these mealworm proteins was slightly higher (~71%) in the case of the mite-specific IgE-negative CA dog sera, while no recognition was observed between such sera of CH dogs and mealworm protein extracts. Cross-reactivity of the 14 kDa protein of the fully digested mealworm extract with the mite-specific IgE-positive CH dog sera was ~62% and ~79% with the corresponding CA canine sera. For the mite-specific IgE-negative CA dog sera, the percentage of the 14 kDa mealworm protein recognition was slightly higher (~86%). Again, no cross-reactivity was observed in the case of mite-specific IgE-negative CH dog sera. Nevertheless, when comparing the two groups of dogs (CH and CA), no statistically significant correlation was found between the cross-reactivity of their sera with the mealworm proteins, clinical signs of allergy and the presence of IgEs against storage mites *T. putrescentiae* and *A. siro* (*p* > 0.05) in their sera.

### 3.5. Identification of Potential Mealworm Protein Allergens

Proteins in the digested acidic and alkaline mealworm extracts were separated by SDS-PAGE, trypsinized and the resulting peptides were identified by LC-MS/MS analysis (Table 3). Altogether, 17 different proteins were identified. Nine of them were found in both extracts, two (larval cuticle protein F1 and cockroach allergen-like protein) only in acidic extract and six (cytochrome P450 monooxygenase CYP4G123, 86 kDa early-stage encapsulation-inducing protein, serpin1, hexamerin 2, alpha-amylase and aldehyde oxidase AOX1) only in alkaline extract. The identified mealworm proteins, the larval cuticle protein, proteins involved in muscle contraction (tropomyosin), can be classified as structural proteins, while α-amylase, glucose dehydrogenase and aldehyde oxidase represent different enzymes.

Allermatch™ was used to verify the allergic potential of the identified proteins. In Allermatch™, a protein can be considered potentially allergenic if it has more than 35% identity to a known allergen within a window of 80 amino acids or more. However, it should be noted that this web tool was developed to predict human food allergens. Nevertheless, 9 of the identified mealworm proteins were defined as allergens in this way (Table 4). Their complete sequences have an identity of 30–96% with known allergens in the Allermatch™ database. The highest sequence identity was observed for the two tropomyosins with allergenic tropomyosins from different insect species. The identity was higher (96%) for the tropomyosin from *T. molitor* (QBM01048) than for that from *Zophobas atratus* (QCI56576) (70–80%), of which only a partial sequence is available. The latter also shares 84% sequence identity with tropomyosin (allergen Aca s 10) from *A. siro* (ABL09305) and 87% with the partial sequence of tropomyosin from *T. putrescentiae* (ABQ96644).

As expected, three mealworm proteins, apolipophorin-III, larval cuticle protein and odorant-binding protein 14, which share very low sequence identity with their counterparts in insect and other Arthropoda species, were not recognized as allergens by Allermatch™. However, their potential allergenicity was recently established as they cross-reacted with IgEs from sera of human patients allergic to shrimp [15].

## 4. Discussion

Insect-based pet food products are already available on the market. However, their suitability as a protein source for dogs has not been extensively studied. Some studies investigated digestibility and immunological parameters in dogs fed on insect-based diet and no adverse effects were found [21,22]. A field study showed that insect protein-based diets can be suggested as an alternative novel protein source for dogs with food intolerances [23]. To our knowledge, this is the first in vitro study trying to establish the link between cross-reactivity of *Tenebrio molitor* proteins with serum IgEs directed to storage mites in dogs with clinical symptoms of allergy.

In a growing number of countries, the larval stages of beetles of the family Tenebrionidae are commercially available as pet feed. According to interviews with dog’s owners, none of the dogs included in our study consumed food containing yellow mealworm proteins; therefore, sensitisation to insect proteins via the oral route was unlikely. Because of the risk of sensitization and the risk posed to existing allergic populations, allergenicity assessment of novel foods is always necessary [24,25]. An important part of the risk assessment of allergenicity of novel food is the characterisation of its protein content [25]. Therefore, in this study, proteins were extracted from mealworm larvae and characterized. Due to the complexity of the insect protein content, various extraction methods described in the literature were tested. Aiming to survey as many mealworm proteins as possible, we performed the extraction from larvae at acidic and at alkaline conditions. As evident from Figure 1, extraction at pH 2 resulted in a larger number of proteins detected on SDS-PAGE gels than the extraction at pH 10.

The allergenic potency of a protein molecule can be predicted from the frequency and intensity of its IgE antibody binding capacity. In line with modern trends in allergology, it is assumed that proteins that are most strongly bound by the patient’s antibodies are considered as the main allergens [26]. Therefore, targeted IgE binding can be used to identify putative allergens. Binding of a particular protein to IgEs from allergic patients indicates that the novel protein can trigger an allergic reaction [27]. In our study, two groups of sera—from clinically healthy (CH) and clinically allergic (CA) dogs were tested for environmental- and feed-specific IgE by ELISA (Table 1). Most ecological studies in temperate climate zones showed that *D. pteronyssinus* (originally known as European HDM) and *D. farinae* (American HDM) are the predominant HDMs worldwide [16]. Despite differences in geographical prevalence, positive reactions to *D. farinae* are most commonly observed in intradermal tests in European dogs [28]. Our clinical experience and research data show that, among environmental allergens, dogs in Slovenia are most frequently sensitized to mites, especially to *D. farinae* and to storage mites [29]. The present study confirmed these findings, with ~77% of dog sera containing IgEs against *D. farinae* and ~71% to the storage mites *T. putrescentiae* and *A. siro*. The sensitivity to both storage mites seems logical due to their phylogenetic proximity. Storage mites have previously been detected both in the dogs’ environment and in their feed [28]. As a result, and due to the lack of data in the literature on sensitivity to storage mites in dogs, we divided dogs into two groups, IgE positive and IgE negative, with respect to the presence or absence of IgEs against the mites *T. putrescentiae* and *A. siro*. The allergenicity of storage mites in humans is well established and several protein allergens were characterized. Some of them can be considered as pan-allergens whose sequence homology and biological function are similar to allergenic proteins in house dust mites *Dermatophagoides* spp. [16].

There is no predictive and validated in vitro method for assessing the allergenicity of novel proteins or protein-containing products [27]. Resistance to denaturation and digestion are important characteristics of many food allergens [24]. Resistance to pepsin is proposed as a criterion for a protein to be considered as a potential allergen [27]. Therefore, in our study a simulated in vitro digestion system with gastric and duodenal enzymes was used to evaluate the digestibility of mealworm proteins. Enzymatic hydrolysis with pepsin, trypsin and α-chymotrypsin was rapid and then the protein digestion profile remained unchanged after overnight incubation (Figure 2). The digests after 18 h were selected as optimal and were used for the Western blot and the LC MS/MS analysis.

In the last decade, LC-MS/MS analysis for identification and characterization of food allergens has become widely used [25]. Using this method, we identified the most abundant proteins in both acidic and alkaline mealworm extracts that resisted extensive simulated gastric and duodenal digestion (Table 3). Most of these proteins, 12 out of 17, have been predicted potential allergens by Allermatch™ algorithm (Table 4) and by available data in the literature. Six identified proteins were detected in both mealworm extracts (tropomyosins, Tm-E1a cuticular protein, odorant-binding protein 14, apolipophorin-III, glucose dehydrogenase), four only in the alkaline extract (alpha-amylase, hexamerin 2, serpin1, 86 kDa early-stage encapsulation inducing protein) and two only in the acidic extract (cockroach allergen-like protein, larval cuticle protein F1). Most relevant for our study are discussed below.

Many authors described cross-reactivity between insects and other Arthropoda (crustaceans, mites) and identified various proteins involved in muscle contraction as pan-allergens [30]. Tropomyosins are highly conserved regulatory proteins involved in the formation of muscular myofibrils and are known as major food allergens as well as respiratory minor allergens from environmental origin (e.g., mites and cockroaches) in humans. Fifteen arthropod tropomyosins have been registered as food allergens. They are strong food allergens and highly cross-reactive within the invertebrate group. Tropomyosin from yellow mealworm larvae has been identified as a major food allergen in humans [31] and possibly also in dogs as shown in our study.

Cuticle proteins characterize the insect external coating and form a complex with chitin, the major constituent of the exoskeleton [30]. Of the two cuticle proteins identified in our study, Tm-E1a cuticular protein (AAB34025) and larval cuticle protein F1 (Q9TXD9), only the former was recognized as an allergen by the Allermatch™ prediction. It shows ~30% sequence identity to the grass pollen allergens Poa p 5 (*Poa pratensis*) and Loa p 5 (*Lolium perenne*). However, larval cuticle protein F1, belongs to a newly identified group of insect allergens consisting of ubiquitous proteins that share well-conserved three-dimensional structures despite low conservation of primary structures. Apolipophorin-III (CDF77373) and odorant-binding protein 14 (AJM71488) also belong to this group of allergens. Molecular modelling of allergenic *T. molitor* proteins apolipophorin-III, larval cuticular protein and hemolymph protein revealed presence of structures similar to allergens occurring in pollens and fruits [15].

Of the proteins identified only in the alkaline mealworm extracts, α-amylase (P56634) is a known allergen in several mite species. Alpha-amylases exhibit a high degree of sequence similarity between mites, insects, and mammals [24], increasing the likelihood that they behave as potentially cross-reacting IgE-binding allergens.

Cockroach allergen-like protein (Q7YZB8) was found in the acidic mealworm extract. It is a nitrile-specifier protein with detoxifying function, which is localized in the microvilar part of the insect’s midgut. Nebbia and colleagues [32] showed that this protein is involved in the primary respiratory and food allergy to mealworms in humans.

Although arginine kinase, an arthropod pan-allergen, was previously identified as an important cross-reactive allergen in the yellow mealworm [24,33], its presence in the digested mealworm protein extracts was not confirmed in our study.

A cross-reactivity study by Verhoeckx et al. [34] suggested that people who are sensitised to HDM and crustaceans may react to the proteins of the yellow mealworm present in food. However, these observations cannot simply be extrapolated to animal patients because the main mite allergens in dogs are different to those in humans [28]. All canine sera that we tested, mite-specific IgE-positive and IgE-negative sera, recognized mealworm proteins in the range 20–30 kDa in Western blot analysis (Figure 3). On the other hand, IgE binding to the mealworm proteins with higher molecular masses (34–55 kDa) and to a 14 kDa protein in the digested acidic extracts was observed in only ~66% of the immunoblots. However, the difference in cross-reactivity was not significant in relation to the clinical status of the dogs or the presence/absence of IgEs against two storage mites in canine sera (*p* > 0.05). The reason for the observed insignificant differences may lay in a relatively small group of animals used in our study. Another possible explanation of such result is also that the IgE testing was not performed in all dogs during an acute allergy phase when concentrations of allergen-specific IgEs are the highest. It is known that the serum half-life of IgEs is only 2 days. Therefore, it is possible that IgEs have already largely disappeared from the circulation of some animals [35].

Recent proteomics and bioinformatics studies have shown that insect allergens such as odorant-binding proteins and apolipophorin III contain a large number of exposed cleavage sites for trypsin and α-chymotrypsin and are therefore readily degraded at alkaline pH [36]. In silico trypsin hydrolysis of other allergens identified in this study predicted good digestibility for all but Tm-E1a cuticular protein, hexamerin 2 and α-amylase. This is supported by the combined results of SDS-PAGE, Western blot and LC-MS/MS analysis and may explain why some proteins were visible on Western blots in mealworm protein extracts but not also in their digests. In the latter, only one protein band of ~14 kDa was detected by canine sera IgEs. Two proteins were identified in this band, Tm-E1a cuticular protein, cockroach allergen-like protein, suggesting that only their antigenic epitopes resisted intense proteolytic degradation. These proteins have been associated with human allergies therefore it is very likely that they are allergens also in dogs and should be further investigated. Tropomyosins, Tm-E1a cuticular protein and larval cuticle protein F1, were identified in the 34–55 kDa protein bands of the digested acidic mealworm extract, consistent with bands detected by all canine sera in Western blot experiments, and where differences in binding were observed, although not significant. Tyr p 10 is an allergen of *T. putrescentiae* that is 64–94% identical to other allergenic tropomyosin and was detected by 12.5% of sera from sensitized human patients [16]. We suggest the allergenic role of tropomyosin and Tm-E1a cuticular protein also in dogs which should be confirmed in the future. It is known that processing (thermal or pressure) and/or hydrolysis (digestion) of food allergen proteins affects the allergenicity of feed. Denaturation can alter IgE-binding epitopes and, thus, reduce binding capacity [8,37]. Normally, smaller peptides are not sufficient to induce IgE-mediated mast cell activation. Therefore, it would also be interesting to investigate whether the processing of mealworm allergens identified in our study affects the allergenicity of insect-based dog feed.

## 5. Conclusions

In view of the expected wider use of insects as a protein source in animal feed in the future, more attention and further research on their allergenic and cross-reacting potential is needed. In our study, conducted to investigate the interaction between *T. molitor* proteins and the immune system in dogs with clinical symptoms of allergy and sensitization to storage mites in comparison to the group of clinically healthy dogs, no distinct correlation was found between sensitisation to storage mites and the clinical status of the dogs. We confirmed the binding of canine serum IgEs to mealworm proteins, but the differences between healthy and allergic dogs were insignificant. In protein extracts of *T. molitor*, we identified several IgE-reactive proteins that were previously shown cross-reactive also to IgEs from sera of humans sensitized to crustaceans or house dust mites. Our results imply that dogs allergic to mites may also clinically show cross-reactivity with mealworm proteins, so caution should be exercised in their case when using yellow mealworm larvae as an alternative protein source.

## Figures and Tables

**Figure 1 animals-11-01942-f001:**
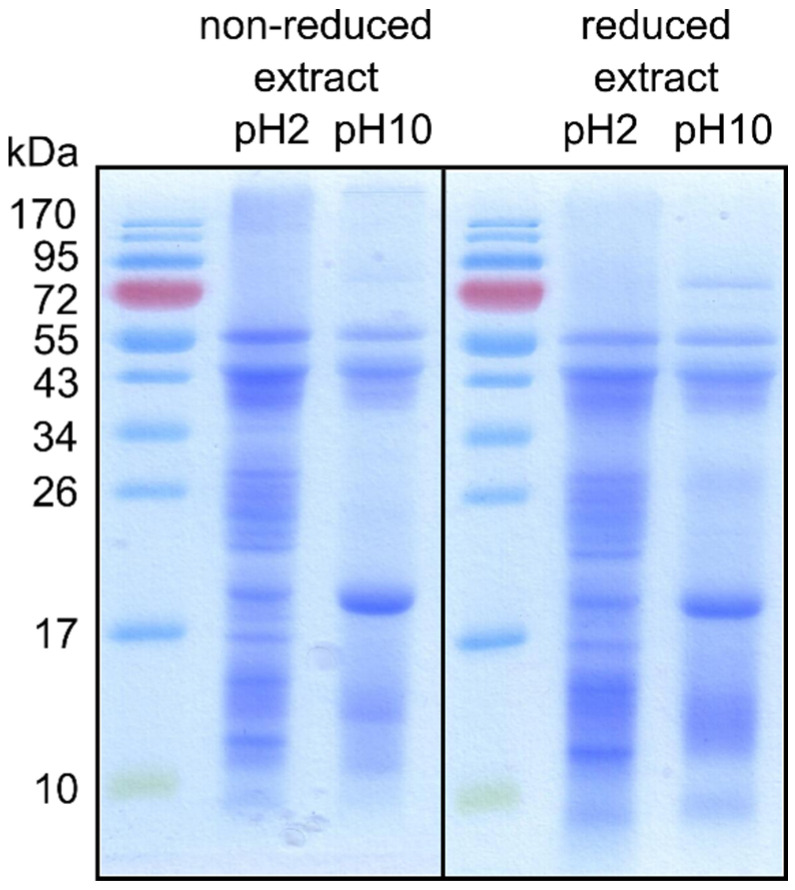
SDS-PAGE analysis of the mealworm protein extracts. The proteins were extracted from dried, ground and defatted larvae under acidic (pH 2) or alkaline (pH 10) conditions and analysed on 12% gel under non-reducing or reducing conditions. Lanes 1 and 4 contain the molecular mass standards. Gels were stained using Coomassie Brilliant Blue dye.

**Figure 2 animals-11-01942-f002:**
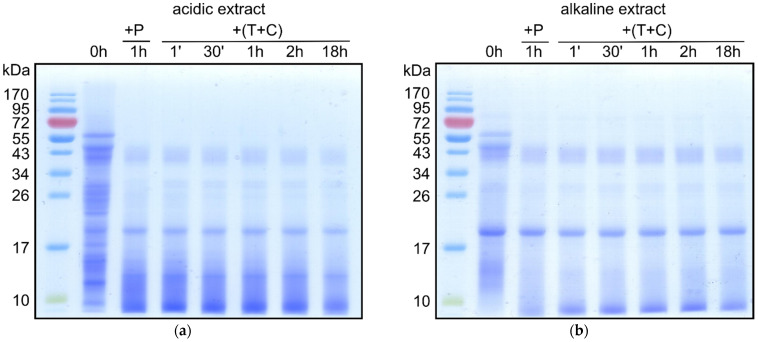
SDS-PAGE analyses of mealworm protein digests. The acidic (**a**) or alkaline (**b**) mealworm protein extract (0 h) was incubated first for 1 h with pepsin (P), and then with trypsin (T) and α-chymotrypsin (C) for different periods of time. Reaction mixtures were electrophoresed on 12% gels under reducing conditions. Lane 1 of each gel contains molecular mass standards. The gels were Coomassie stained.

**Figure 3 animals-11-01942-f003:**
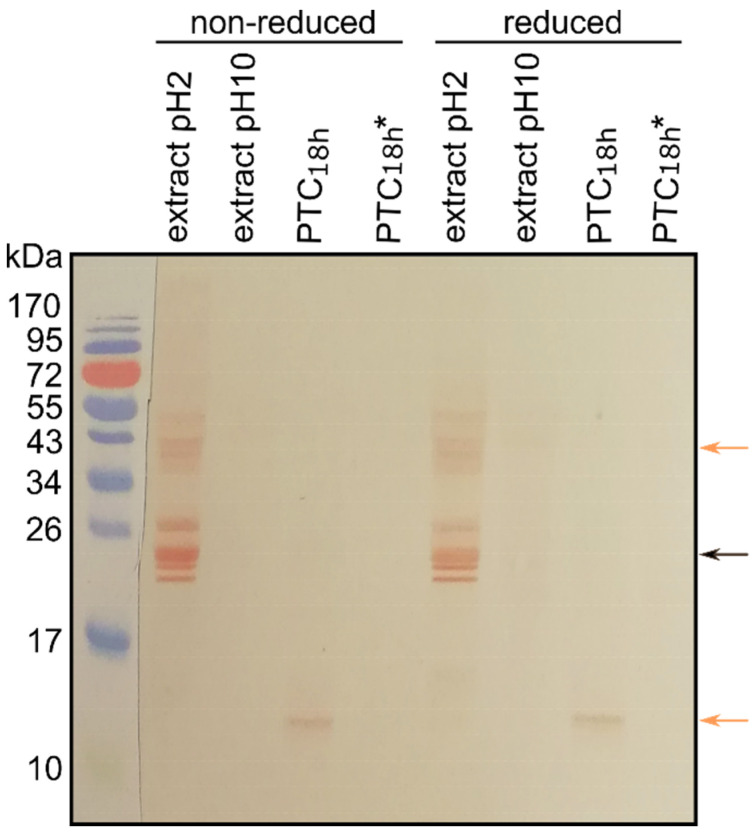
Cross-reactivity of mealworm protein extracts and the serum of a clinically allergic dog. The representative immunoblot shows cross-reactivity between CA canine serum and undigested (extraction at pH 2 or pH 10) and in vitro fully digested mealworm protein acidic (PTC_18h_) or alkaline (PTC_18h*_) extracts. Orange arrows point to protein bands where differences between sera from CH and CA dogs were observed. Black arrow points to the group of mealworm proteins detected by all canine sera.

**Table 1 animals-11-01942-t001:** The presence of mite-specific IgE antibodies in sera of clinically healthy (CH) and clinically allergic (CA) dogs.

Mite	Positive Serum	Negative Serum	Borderline Serum
*A. siro*			
CH (*n* = 10)	8	1	1
CA (*n* = 21)	14	7	0
*T. putrescentiae*			
CH (*n* = 10)	8	1	1
CA (*n* = 21)	14	7	0
*D. farinae*			
CH (*n* = 10)	7	1	2
CA (*n* = 21)	17	4	0
*D. pteronyssinus*			
CH (*n* = 10)	1	9	0
CA (*n* = 21)	1	20	0

**Table 2 animals-11-01942-t002:** Correlation between clinical signs of allergy in dogs, the presence of anti-mite IgEs in their sera and the cross-reactivity of the sera with proteins in mealworm protein acidic extract. Mite-specific IgEs (*T. putrescentiae* and *A. siro*) were determined with ELISA testing in sera of dogs, clinically healthy (CH) or allergic (CA) ones. One serum with borderline values of anti-mite IgEs in the CH group was not included in the study. In the Western blot experiment, canine sera were then probed for the cross-reactivity with proteins extracted from mealworms at pH 2, before exposure to digestive enzymes and after.

Cross-Reactivity with Mealworm Proteins	CH Dogs (*n* = 10)			CA Dogs (*n* = 21)		
	IgE-Positive (*n* = 8)	IgE-Negative (*n* = 1)	∑	IgE-Positive (*n* = 14)	IgE-Negative (*n* = 7)	∑
Before digestion (34–55 kDa)	6	0	6	8	5	13
After digestion (14 kDa)	5	0	5	11	6	17

**Table 3 animals-11-01942-t003:** List of proteins identified by LC-MS/MS in the digests of acidic and alkaline extracts of proteins of yellow mealworm larvae.

Protein Extract	Identified PROTEIN	NCBI Accession	S. Mill Score	Distinct Peptides	Seq. Coverage	Matched Peptide Sequences	Theoretical Mass	pI
pH 2	tropomyosin [*Tenebrio molitor*]	QBM01048	55.92	5	17	LAEASQAADESFR LAEASQAADESFRMCK SQQDEERMDQLTNQLK SQQDEERMDQLTNQLKEAR TLTNAESEMAALNR	32,428.5	4.8
	Tm-E1a cuticular protein [*Tenebrio molitor*]	AAB34025	54.65	3	20	VASPAVSVHPAPAVR YAAPAVASVGYAAPALR YAAPAVASVGYAAPAVR	23,188.5	9.54
	odorant-binding protein 14 [*Tenebrio molitor*]	AJM71488	49.73	5	35	KTGVATEAGDTNVEVLK ATPEETAYDTFK LKHVASDEEVDKIVQK TGVATEAGDTNVEVLK TGVATEAGDTNVEVLKAK	14,706.1	7.58
	Larval cuticle protein F1 [*Tenebrio molitor*]	Q9TXD9	43.71	4	50	GLIGAPIAAPIAAPLATSVVSTR SLYGGYGSGLGIAR STPGGYGSGLIGGAYGSGLIGGGLYGAR YGLGAPALGHGLIGGAHLY	14,566.8	9.63
	28 kDa desiccation stress protein [*Tenebrio molitor*]	AAB41285	41.31	3	26	HKETIPSKTEICSTATSLR TKNVALGVFDALVAPCSHINEV VVDDCLPDSAKGLPSLGVK	24,833.7	5.37
	apolipophorin-III [*Tenebrio molitor*]	CDF77373	24.52	2	9	NLDDGLKTAVAQVEK NLDDGLKTAVAQVEKLVK	21,106.3	8.63
	56 kDa early-stage encapsulation-inducing protein [*Tenebrio molitor*]	BAA78480	21.59	1	2	GVPQYTVGQYGIPR	62,446	8.33
	tropomyosin, partial [*Zophobas atratus*]	QCI56576	20.18	2	22	EVDRLEDELVAEKER FLAEEADKKYDEVAR	15,572.4	4.58
	cockroach allergen-like protein [*Tenebrio molitor*]	Q7YZB8	11.74	1	2	ALDEVQTLAQR	65,481.44	4.08
	nero, partial [*Cryphaeus* sp. INB181]	AUW69182	6.32	1	9	NIQSKEAIEALGAGLK	18,852.6	4.7
	glucose dehydrogenase, partial [*Cryphaeus* sp. INB181]	AUW87486	3.17	1	2	IRRGSR	33,739.6	10.23
pH 10	odorant-binding protein 14 [*Tenebrio molitor*]	AJM71488	136.86	8	60	ATPEETAYDTFK KTGVATEAGDTNVEVLK KTGVATEAGDTNVEVLKAK LKHVASDEEVDKIVQK TGVATEAGDTNVEVLK TGVATEAGDTNVEVLKAK CIYDSKPDFSPID ISKECQQVSGVSQETIDKVR	14,706.1	7.58
	hexamerin 2 [*Tenebrio molitor*]	AAK77560	90.69	5	10	GGMTYQFYVMVSK HLLGYSQQPLTYFK HYYNEHDLMYQGVEVK NVAAYSKPEVVEQFYK YDNRGEAFYYMYQQILAR	84,543.1	6.18
	apolipophorin-III [*Tenebrio molitor*]	CDF77373	68.15	5	22	LSQTAAQLQQAAGPEATAK LSQTAAQLQQAAGPEATAKAK NLDDGLKTAVAQVEK NLDDGLKTAVAQVEKLVK QVQEKLSQTAAQLQQAAGPEATAK	21,106.3	8.63
	56 kDa early-stage encapsulation-inducing protein [*Tenebrio molitor*]	BAA78480	59.91	4	11	EYQGVVDEAQYEK GVPQYTVGQYGIPR GVQTIGQLR GVQTIGQLRQYYPTSLNVNPLLGR	62,446	8.33
	tropomyosin [*Tenebrio molitor*]	QBM01048	55.8	6	24	KLAFVEDELEVAEDRVK LAEASQAADESFR MQAMK SQQDEERMDQLTNQLK SQQDEERMDQLTNQLKEAR TLTNAESEMAALNR	32,428.5	4.8
	alpha-amylase [*Tenebrio molitor*]	P56634	40.72	3	9	GVLIDYMNHMIDLGVAGFR HMSPGDLSVIFSGLK NSIVHLFEWK	51,240.7	4.53
	aldehyde oxidase AOX1 [*Tenebrio molitor*]	AKZ17716	27.91	1	4	IYTIEGIGDPLTGYHPVQEVLAK	138,948.4	5.06
	Tm-E1a cuticular protein [*Tenebrio molitor*]	AAB34025	23.51	2	13	VASPAVSVHPAPAVR YAAPAVASVGYAAPAVR	23,188.5	9.54
	tropomyosin, partial [*Zophobas atratus*]	QCI56576	22	2	22	EVDRLEDELVAEKER FLAEEADKKYDEVAR	15,572.4	4.58
	cytochrome P450 momooxigenase CYP4G123	AKZ17712	19.81	1	3	GIRGSTAEVPVELQTK	58,314.8	8.99
	86 kDa early-stage encapsulation-inducing protein [*Tenebrio molitor*]	BAA81665	19.21	2	4	HLLGYAYQPYTYHK VYVDANTETDAVVK	90,623.9	6.62
	nero, partial [*Cryphaeus* sp. INB181]	AUW69182	11.67	1	9	NIQSKEAIEALGAGLK	18,852.6	4.7
	28 kDa desiccation stress protein [*Tenebrio molitor*]	AAB41285	11.13	1	8	VVDDCLPDSAKGLPSLGVK	24,833.7	5.37
	serpin1 [*Tenebrio molitor*]	BAI59109	39.76	3	12	AEFLELPFKGNEASMMIVLPK AVLINALYFK TALHLPDDKETVESAIK	44,213.9	6.17
	glucose dehydrogenase, partial [*Cryphaeus* sp. INB181]	AUW87486	3.17	1	2	IRRGSR	33,739.6	10.23

**Table 4 animals-11-01942-t004:** Identified allergenic proteins of the yellow mealworm according to Allermatch^TM^. For the proteins with the large number of matches, only the three best matches found are listed.

	Sequence Identity in Allermatch					
		80 AA Sliding Window Analysis		Full Sequence Alignement		
Protein	Allergen	# Hits > 35% Identity	% hits > 35% Identity	%	Overlap (AA)	E
tropomyosin (QCI56576)	Chi k 10 [*Chironomus kiiensis*]	55	100	96.3	134	1.3^−31^
	Aed a 10 [*Aedes aegypti*]	55	100	95.5	134	2.1^−31^
	Lep s 1.0102 [*Lepisma saccharina*]	55	100	96.7	90	1.1^−19^
tropomyosin (QBM01048)	Lep s 1.0101 [*Lepisma saccharina*]	205	100	82.4	284	1.5^−94^
	Aed a 10 [*Aedes aegypti*]	205	100	77.1	284	1.1^−87^
	Eup p 1 [*Euphausia pacifica*]	205	100	68.3	284	2.0^−76^
alpha-amylase (P56634)	Bla g 11 [*Blatella germanica*]	392	100	64.6	480	5.0^−143^
	Per a 11 [*Periplaneta americana*]	392	100	58.9	477	2.1^−125^
	Der f 4.0101 [*Dermatophagoides farinae*]	392	100	49.5	497	4.1^−69^
hexamerin 2 (AAK77560)	Per a 3 [*Periplaneta americana*]	397	63.72	39.1	665	7.7^−46^
	Bla g 3 [*Blatella germanica*]	403	64.69	38.0	681	3.1^−46^
86 kDa early-stage encapsulation-inducing protein (BAA81665)	Per a 3 [*Periplaneta americana*]	431	63.85	37.9	702	1.8^−44^
	Bla g 3 [*Blatella germanica*]	392	58.07	37.1	672	1.6^−41^
cockroach allergen-like protein (Q7YZB8)	Bla g 1 [*Blatella germanica*]	340	65.89	35.9	412	1.5^−42^
	Per a 1 [*Periplaneta americana*]	259	50.15	35.6	413	4.0^−40^
glucose dehydrogenase (AUW87486)	Mala s 12 [*Malassezia sympodialis*]	69	31.22	32.4	278	6.7^−21^
serpin1 (BAI59109)	Tri a 33 [*Triticum aestivum*]	89	28.25	32.0	369	1.0^−28^
	Gal d 2.0101 [*Gallus gallus*]	14	4.4	28.1	388	1.7^−27^
Tm-E1a cuticular protein (AAB34025)	Poa p 5 [*Poa pratensis*]	40	24.39	29.6	226	0.11
	Lol p 5 [*Lolium perenne*]	78	47.56	28.3	198	9.0^−07^
	Poa p 5 [*Poa pratensis*]	61	37.20	27.2	246	0.97

## Data Availability

Not applicable.

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
