# Peer review of "Insect Protein-Based Diet as Potential Risk of Allergy in Dogs"

_animals, 2021, doi:10.3390/ani11071942_

Round 1

Reviewer 1 Report

Bajuk and colleagues have described an interesting work on the potential risk of allergies in dogs fed with insect proteins, however there are two main major revisions to justify.

"Thirty-one client-owned dogs of different age and breeds were included in the study."

Thank you rof being helpfull.

Specifying race, age and gender can be helpful for readers. Please clarify.

"dogs were divided into two 123 groups: clinically healthy (CH; N=10) and clinically allergic (CA; N=21)".

What are the reasons that led to such a marked difference in numbers in the two groups?

Reviewer 2 Report

The authors studied the interaction between T. molitor proteins and IgE of healthy and clinically allergic dogs.
The methods are sophisticated and the work shades light on a subject of controversy in veterinary allergology, the existence and the manifestations of IgE-mediated food allergy. 
It could be interesting to describe the clinical manifestations of IgE-mediated food allergy in humans, i.e. anaphylaxis (lane 73 - 74).

To clarify the study, the authors could give the international definition of food hypersensitivity and food allergy : Johansson et al. J Allergy Clin Immunol 2004.

So, in the dog, IgE-mediated food allergy would probably induce anaphylaxis as shown in two articles : Rostaher et al. Vet Dermatol 2017;28:251-e66 and Kang and Park. Can Vet J 2012;53:1203-6.

Do the clinically allergic dogs of this study have manifestations compatible with true IgE-mediated food allergy?

They have, more likely, food hypersensitivities, with sensitization to various non pertinent food allergens and this could explain the non statistically significant correlations observed. The authors could discuss this point.

Reviewer 3 Report

Dear authors,

I consider your work a great contribution to the field of small animals internal medicine.

I suggest, however, some reflections.

Introduction

Line 52 - It would be interesting to refer to the notion of protein of high biological value.

Line 67 - essential ones? (aa)

Lines 75 - 76 Food allergies commonly cause skin, respiratory, gastroenterological and ear clinical signs in dogs and cats.

Reference (https://www.dvm360.com/view/diagnosing-food-allergies-dogs-and-cats-bring-your-case-trial): "Clinical signs in dogs: Clinical signs include nonseasonal pruritus, otitis, dermatitis, eosinophilic vasculitis, recurring pyoderma, seborrhea or urticaria. Nearly half of my food allergy patients have gastrointestinal signs. These signs may include vomiting, diarrhea, flatulence or more than two bowel movements a day. Rarely reported clinical signs of adverse food reactions include seizures and respiratory signs, including bronchitis, rhinitis and chronic obstructive pulmonary disease.1"

Results

Line 318 - Table 2. Correlation between clinical signs of allergy in dogs, the presence of anti-mite IgEs in their

Discussion

In the discussion, it would be interesting to mention a safe alternative for the use of these resources, which is the treatment by hydrolysis of proteins that will decrease the activation of IgE and the degranulation of mastocytes and the consequent inflammatory / allergic phenomenon.

Lines 384-385 A field study showed that insect protein-based diets can be suggested as an alternative feed source for dogs with food intolerances as a novel protein [20].
